# A dose scaling antivenin protocol in treatment of *Daboia palaestinae* envenomation may reduce morbidity and costs

Daniel J. Jakobson[1,2], Zurab Zakariashvili[3], Enzo F. Galicia H.[3], Mohammad Abu Issa[1], Miguel M. Glatstein[4,5], Frederic S. Zimmerman[1¤*]

**1** Intensive Care Department, Barzilai University Medical Center, Ashkelon, Israel, **2** Faculty of Health Sciences, Ben-Gurion University of the Negev, Beer-Sheba, Israel, **3** Department of Anesthesiology, Barzilai University Medical Center, Ashkelon, Israel, **4** Division of Pediatric Emergency Medicine, Dana-Dwek Children's Hospital, School of Medicine, Tel Aviv University, Tel Aviv, Israel, **5** Division of Clinical Pharmacology and Toxicology, Ichilov Hospital, Tel Aviv University, Tel Aviv, Israel

¤ Current address: Critical Care Unit, Shaare Zedek Medical Center, affiliated with the Hebrew University-Hadassah Medical School, Jerusalem, Israel
* fzimmer@szmc.org.il

## Abstract

### Background

*Daboia palaestinae* is a leading cause of snakebite envenomation in the eastern Mediterranean, with substantial mortality in the absence of antivenin. Current recommended antivenin dose is 50 ml; however, antivenin is costly, may be difficult to obtain and is associated with substantial side effects. Thus, this study was designed to define the minimal effective antivenin dose and identify patients who can be safely managed without antivenin.

### Methods

This retrospective single-center study was conducted in adults with suspected or confirmed D. *palaestinae* envenomation. Patients were treated via our previously developed envenomation protocol: no antivenin use for local symptoms and dose scaling for mild or severe systemic symptoms – initially 10 ml antivenin, with repeat dosing for ongoing systemic symptoms. The main outcomes measured were morbidity and mortality associated with this protocol. Secondary outcomes included assessing the demographics and clinical effects of snake envenomation and comparing between those who received antivenin and those who did not.

### Results

In total, 101 patients were included. A median of 45 minutes [interquartile range: 30–61 minutes] elapsed between snakebite and hospital admission, with no differences between groups. Among 52 patients receiving antivenin, 119 [60–237] minutes

**Data availability statement:** All data files are available from the Open Science Foundation database (accession number osf.io/2p685).

**Funding:** The author(s) received no specific funding for this work.

**Competing interests:** The authors have declared that no competing interests exist.

elapsed between snakebite and initial antivenin administration, with a maximum of 1073 minutes to initial antivenin administration. Maximum until last antivenin was 3860 minutes.

Median antivenin dose was 15 [10–22.5] ml, with 26/52 (50.0%) requiring only 10 ml. Two (2) patients developed an early antivenin immune reaction, with one developing anaphylaxis requiring invasive ventilation. Both received a single 10 ml dose of antivenin prior to allergic reaction. Neither patient had a known history of exposure to serum or relevant allergic reaction. No patients died during hospitalization.

## Conclusions

This cohort demonstrates that a dose-scaling antivenin protocol can be safely employed, reducing morbidity and costs. We recommend a randomized control trial comparing fixed dose regimen to an escalation protocol and development of similar protocols for envenomations due to other snake species.

---

## Introduction

Snake envenomation is a widespread cause of mortality and morbidity worldwide, with snakebites resulting in 20–130,000 worldwide deaths annually. Native species of venomous snakes are found on all continents except Antarctica and some island countries, whereas non-native snakes may also be encountered as pets [1–3]. Clinical signs and symptoms, as well as symptom severity, differ between and within species, and are dependent on the chemical composition of the venom, which varies substantially between species, with intraspecies variation occurring as well, depending on factors including gender, age, prey availability, diet and geographic location, among others [2,4].

*Daboia palaestinae (*previously *Vipera palaestinae* or *Vipera xanthina palaestinae*) is a viper species endemic to the eastern coasts of the Mediterranean, especially the coastal plains and inland hills of Israel and Lebanon, and is the most common venomous snake and the leading causing of snakebite envenomation in this region [4,5], with 100–300 envenomations due to bites by this species from Israel annually [6,7]. Taxonomically, it is from the Viperinae subfamily, which is distributed widely across Africa, Europe, the Middle East and Asia, including the islands of the far east [4].

The venom of this snake primarily contains myotoxins, procoagulants and hemorrhagins, as well as neurotoxins, angioneurin growth factors, and integrin inhibitors which can cause both local and systemic effects [4–6,8]. Local effects include severe pain and swelling of the affected limb, whereas systemic affects include nausea, vomiting and abdominal pain, with more severe envenomizations resulting in respiratory distress, hypotension and convulsions, up to hypovolemic and vasodilatory shock which, left untreated, can lead to cardiac arrest. Neurological symptoms, coagulopathy, thrombocytopenia and necrosis have also been reported [5,9–11]. Effects may appear immediately after snakebite, but are often delayed, including beyond 24 hours [10].

The primary and most effective treatment of snakebite envenomation, including that resulting from *D. palaestinae* bite*,* is administration of specific or polyvalent snake antivenin, use of which drastically reduces mortality, with a reduction in morbidity as well [9,10]. Currently in Israel, including in our institution during the study period, a monovalent whole immunoglobulin *D. palaestinae* antivenin (Kamada, Beit-Kama, Israel) is administered with a fixed dose regimen of 50 ml of antivenin is recommended in all *D. palaestinae* envenomations, including both systemic manifestations and progressive local manifestations of envenomation where *D. palaestinae* bite is suspected or confirmed. This protocol was developed due to the feeling that a lesser dose is ineffective, though no high-quality data is available [5,9,11,12]. However, antivenin administration is costly, with a single fixed dose regimen for treating *D. palaestinae* in Israel estimated to cost $7000, with repeat administration sometimes required [5,6,13]; antivenin costs for other species in other locales vary significantly but remain a substantial financial burden for most health care systems [14]. Furthermore, certain antivenins have historically been in short supply. These limitations can form substantial barriers to treatment access, especially in limited-resource settings. Moreover, antivenin administration can itself be associated with substantial morbidity, including anaphylaxis, chills, fever, pain at injection site and serum sickness, including delayed reactions up to a month after administration [5,9,11]. Thus, though antivenin administration is absolutely essential in preventing morbidity and mortality in the relevant patient population, it is also important to define the minimal necessary dose to effectively treat envenomation, as well as define patient populations who can be safely treated without antivenin administration. The achievement of such objectives will minimize antivenin side effects, reduce costs and help to increase the availability of antivenin for those who truly need it.

In contrast to this fixed-dose regimen, we previously developed an envenomation treatment protocol that minimized use of antivenin in order to minimize side effects and overall morbidity, as well as reduce antivenin use in periods of scarcity [11]. This protocol involves conservative treatment of local symptoms without use of antivenin [6,7,9] and treatment of both mild and severe systemic symptoms employing dose scaling antivenin, with an initial 10 ml dose of antivenin, with repeat dosing in patients with ongoing systemic symptoms, in accordance with treatment response. This protocol has the potential to reduce antivenin morbidity as well as substantially reduce costs and increase antivenin availability [11] and is consistent with similar protocols developed for snake envenomations from other species [15,16]. It should be noted that since initial publication of this escalation protocol, quality control of antivenin production has improved, possibly contributing to reduced morbidity from antivenin administration [1,5,13]. Furthermore, we have subsequently further reduced initial antivenin dosing relative to the initial publication. Thus, the primary aim of the current study is to describe the outcomes of a snake envenomation treatment using an escalated dose antivenin protocol in terms of length of stay, morbidity and mortality. Secondary aims include assessing the demographics and clinical effects of snake envenomation and comparing between those who received antivenin and those who did not.

## Materials and methods

### Study site

This retrospective single-center study was conducted at Barzilai University Medical Center (BMC), a 500-bed hospital in Ashkelon, Israel. Israel is located in the eastern Mediterranean Basin and is characterized by large variety of habitats including Mt. Hermon – 2200 m above sea level with annual snow, the Dead Sea area – 400 m below sea level with Afro-tropical flora and fauna and a Saharo-Arabian eremic zone in the far south. In contrast, the north and center of the country consist of a Mediterranean habitat and in the south and east Irano-Turanian grassland and deserts are found. The city of Ashkelon sits on the coastline and its environs are characterized by the latter two habitats [17]. These are also the habitats of *Daboia palaestinae* (Fig 1) [18]. In fact, though nine venomous snakes are identified regularly throughout the various habitats of Israel, only *Daboia palaestinae* is found in the environs of Ashkelon, though numerous non-venomous snakes do occupy the same habitat [19,20]. Thus, *Daboia palaestinae* snakebite is most common cause of snakebite in our institution, and practically the only cause of snakebite in victims showing signs of envenomation.

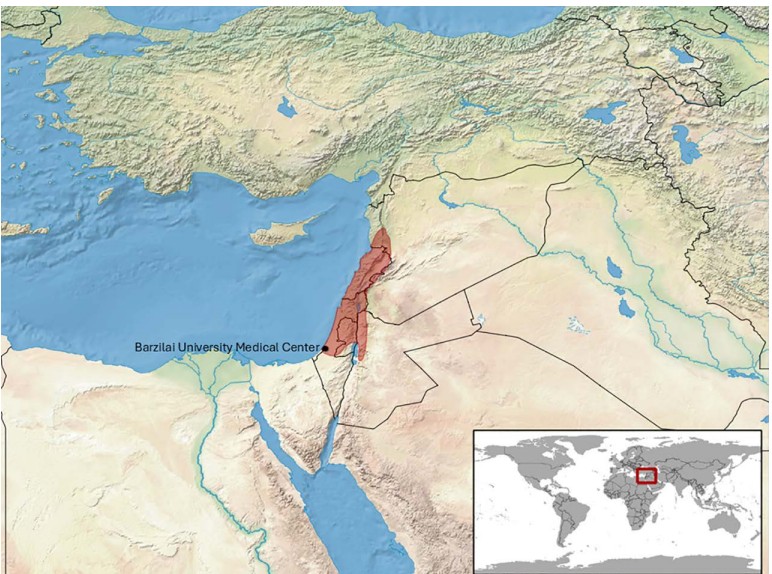

**Fig 1.  *Daboia palaestinae* distribution.** Adapted from "*Daboia palaestinae* distribution" by IUCN Red List of Threatened Species, used under CC BY 3.0.

## Study subjects

Study subjects included all adult patients hospitalized in BMC from 2014−2023 with an admission diagnosis of snakebite (ICD-10 code T63.0) in which D. *palaestinae* envenomation was suspected or confirmed. Diagnosis of *D. palaestinae envenomation* was made by the treating clinician by clinical symptoms, species description and, when available, by identification of the species either directly or by photograph. All patients were preferentially initially admitted to the intensive care unit, a 12-bed mixed ICU with both medical and surgical patients.

## Inclusion/exclusion criteria

Patients admitted with a diagnosis of snakebite in which D. *palaestinae* envenomation was ruled out were excluded. Thus, of the 114 patients were admitted to BMC with a snakebite diagnosis, during the study period, 10 patients were eliminated after ruling out *D. palaestinae* envenomation, two more patients were eliminated due to initial treatment in another hospital and one minor patient was eliminated. Accordingly, 101 patients were included in the current study. Of these, 49 patients did not receive antivenin and 52 received antivenin (Fig 2).

Admitted patients were preferentially treated according to the protocol currently being applied in our medical center. Thus, patients exhibiting systemic signs of envenomation – including, but not limited to, nausea, vomiting, abdominal pain, diarrhea, blood pressure changes, arrhythmias, and anaphylactic shock – were immediately administered 10 ml of antivenin. Further doses of 10 ml of antivenin were administered if abatement of symptoms was not noted within a short period of time or if symptoms recurred. Patients with local symptoms and signs only did not receive antivenin. All patients were kept under observation in the intensive care unit until abatement of systemic symptoms and for at least 48 hours due to risk of delayed symptoms

## Data access

Data were accessed for research on September 12, 2023 and included date and time of envenomation, age, sex, time of presentation to the hospital, clinical signs and symptoms as well as laboratory results on presentation and during

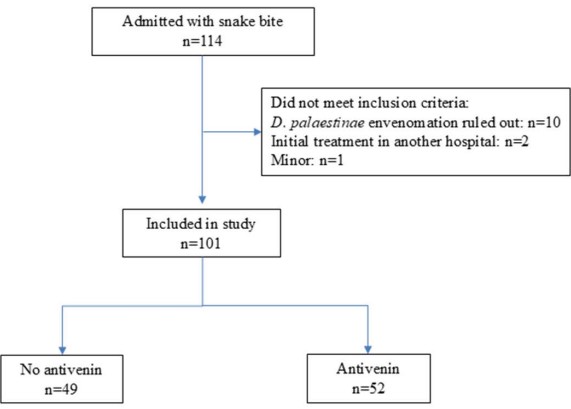

**Fig 2. Inclusion/ exclusion criteria.**

hospitalization, time and dosage of antivenin administration, need for repeat administration, complications and complication management, as well as need for adjunctive therapy. Length of stay in the intensive care unit and in-hospital was also recorded. Symptoms and signs were classified as local (confined to the limb affected, e.g., swelling, pain, discoloration), mild systemic (e.g., gastrointestinal symptoms) and severe systemic (e.g., neurological or cardiovascular compromise). All subject identifiers were removed by subject coding prior to data analysis, thus authors had access to data individual participant information during data collection but not after collection and during analysis.

## Study outcomes

The primary outcome of this study was in-hospital morbidity and mortality associated with an escalated dose antivenin protocol as measured by clinical signs and symptoms as well as laboratory results. Secondary outcomes included evaluation and description of the demographics and clinical effects of snake envenomation, including comparisons between those who received antivenin and those who did not.

## Data analysis

The dataset includes all cases of *Vipera palaestinae* envenomation treated during 2014–2023, representing a consecutive, non-randomized case series. Statistical analyses were performed using R (Version 4.2.2. R Foundation for Statistical Computing. released 2022. Vienna, Austria). Descriptive statistics (i.e., numbers, proportion and means) were used to describe the study population. A secondary analysis compared subjects with local symptoms and signs only –not treated with antivenin by our protocol – to those with systemic symptoms or signs. Categorical variables (e.g., sex, chronic illness, location of envenomation, symptoms, use of supplemental treatments) were analyzed busing the Chi-square test or Fisher's exact test when expected frequencies were less than five. Normality tests were performed (Shapiro-Wilk) on continuous variables to determine the appropriate statistical tests. Continuous variables with normal distribution (e.g., age, laboratory values) were analyzed using Student's t-test, while non-normally distributed variables (e.g., admission times, length of ICU and hospital stay) were analyzed using the Mann–Whitney U test. In all tests, two-tailed p-values were taken and a p-value <0.05 was considered significant.

## Ethical considerations

The study was approved by the BMC institutional review board prior to initiation (approval number: 0009–23-BRZ). Due to the retrospective nature of the study, informed consent was waived.

## Results

101 patients were included in the current study – 49 did not receive antivenin and 52 did. The average age of the study population was 41±5 years. 78/101 (77.2%) of patients were male. 24/101 (23.8%) had had a chronic condition. No demographic differences were noted between those who did and did not receive antivenin (Table 1).

Envenomation primarily occurred between May and November, with a small number of envenomations in the off-season and a peak at the beginning and towards the end of the season. It should be noted that the peak towards the end of the season primarily involved envenomations not requiring antivenin treatment (Fig 3).

Envenomation occurred in the upper limb in 77 of 101 patients (76.2%), in the lower limb in 41 patients (40.6%) with one patient receiving a bite on the scalp. 18 patients received snakebites in multiple locations. 67 of 101 patients (66.3%) experienced pain at the location of the bite and 100 of 101 (99.0%) patients experienced local swelling. No differences were noted in bite location or local symptoms between those who received antivenin and those who did not (Table 1).

On aggregate, 57 of the 101 patients (56.4%) experienced one or more generalized symptoms. 51 of these received antivenin as per institutional protocol. Seven protocol deviations occurred, in accordance with clinical judgement of the treating physician, with one patient without generalized symptoms receiving antivenin and six patients with mild generalized symptoms not receiving antivenin (Table 1). After chart review, the latter deviations were due to clinical judgement attributing symptoms to treatment, rather than to envenomation.

White blood cell count (WBC) on admission of the antivenin group was increased versus the non-antivenin group – $9.2\pm0.2$ x $10^3/\mu l$ versus $7.7\pm0.2$ x $10^3/\mu l$ (p=0.008). Platelet and creatinine levels on admission were also increased in the antivenin versus the non-antivenin group – $246\pm165$ x $10^3/\mu l$ versus $225\pm147$ x $10^3/\mu l$ platelets and $1.00\pm0.00$ and $0.95\pm0.00$ creatinine. However, this did not reach statistical significance (p=0.092 and p=0.056, respectively). Mean international normalized ratio (INR) and mean fibrinogen for the entire population on admission were $1.14\pm0.01$ and $259\pm177$, respectively, with no significant differences noted between groups.

Median time between snakebite and hospital admission was 45 [30–61] minutes, with 121 [81–179] minutes from hospital to ICU admission and 183 [127–260] minutes between snakebite and ICU admission. No differences in admission times were noted between groups. In the antivenin group, median time between snakebite and initial antivenin treatment was 119 [60–237] minutes, with a median of 48 [24–222] minutes elapsing between hospital admission and antivenin treatment. Median time from snakebite to last antivenin treatment was 195 [110–324] minutes (Table 1). Some patients developed symptoms late, thus the maximum time between snakebite and antivenin administration was 1073 minutes. Other patients developed recurrent symptoms despite early antivenin administration; thus, the maximum time between snakebite and last antivenin administration was 3860 minutes.

During the course of their hospitalization, patients who were treated with antivenin received a median of 15 [10–22.5] ml of antivenin in a median of one [1–2] administration (Table 2). 26/52 (50.0%) patients receiving antivenin required only 10 ml antivenin (one dose according to institutional protocol), with 13/52 (25.0%) requiring 20 ml and 13/52 (25.0%) requiring more than 20 ml. The maximum dose administered was 70 ml, which was received by one patient in three separate administrations. Two patients developed an early immune reaction to antivenin, with one of these developing anaphylaxes requiring invasive ventilation.

Besides antivenin, 7/101 patients (6.9%) required vasoactive medication, five (5.0%) required supplemental oxygen and, of these, two (2.0%) required invasive ventilation. Two (2.0%) required blood products. All these were in the group that received antivenin. Additionally, 15/52 (28.8%) in the antivenin group and 4/49 (8.2%) in the non-antivenin group, were treated with antiemetics (p=0.010), 35/52 (67.3%) and 15/49 (30.6%) – respectively – required non-opiate analgesia (p<0.001), 28/52 (53.8%) and 13/49 (26.5%) required opiate analgesia (p=0.008) and 3/52 (5.8%) and 1/49 (2.0%) received surgical intervention (p=0.618); surgical intervention was carried out in deviation from standard protocol [1,21].

In terms of laboratory values, mildly increased maximal leukocytes were noted in the antivenin group relative to the non-antivenin group (median 11.1 [9.2–13.4] x $10^3$ per $\mu l$ versus 8.4 [7.0–10.1] x $10^3$ per $\mu l$, p<0.001). A reduction in

**Table 1. Demographics and presentation.**

| Variable | All patients n=101 (%) | Antivenin n=52 (%) | No antivenin n=49 (%) | p-value |
|---|---|---|---|---|
| Male | 78 (77.2) | 37 (71.2) | 41 (83.7) | 0.159 |
| Age – years ±CI | 41±5 | 41±7 | 41±6 | 0.793 |
| Any chronic illness | 24 (23.8) | 12 (23.1) | 12 (24.5) | 1.000 |
| Diabetes | 3 (3.0) | 1 (1.9) | 2 (4.1) | 0.610 |
| Hypertension | 13 (12.9) | 7 (13.5) | 6 (12.2) | 1.000 |
| Ischemic heart disease | 3 (3.0) | 1 (1.9) | 2 (4.1) | 0.610 |
| Neurological disorder | 3 (3.0) | 1 (1.9) | 2 (4.1) | 0.610 |
| Psychiatric disorder | 1 (1.0) | 1 (1.9) | 0 (0.0) | 1.000 |
| Other | 15 (14.9) | 8 (15.4) | 7 (14.3) | 1.000 |
| Location of bite | | | | |
| Upper limb | 77 (76.2) | 36 (69.2) | 41 (83.7) | 0.105 |
| Lower limb | 41 (40.6) | 26 (50.0) | 15 (30.6) | 0.068 |
| Other | 1 (1.0) | 1 (1.9) | 0 (0.0) | 1.000 |
| Local symptoms | | | | |
| Pain | 67 (66.3) | 39 (75.0) | 28 (57.1) | 0.063 |
| Swelling | 100 (99.0) | 52 (100.0) | 48 (98.0) | 1.000 |
| # of large joints involved ±CI | 2±0 | 2±0 | 2±0 | 0.324 |
| Any generalized symptom | 57 (56.4) | 51 (98.1) | 6 (12.2) | 0.000 |
| Reduced consciousness | 1 (1.0) | 1 (1.9) | 0 (0.0) | 1.000 |
| Sweating | 10 (9.9) | 10 (19.2) | 0 (0.0) | 0.001 |
| Hypotension | 19 (18.8) | 19 (36.5) | 0 (0.0) | 0.000 |
| Arrythmia | 8 (7.9) | 8 (15.4) | 0 (0.0) | 0.006 |
| Shortness of breath | 7 (6.9) | 6 (11.5) | 1 (2.0) | 0.113 |
| Desaturation | 3 (3.0) | 3 (5.8) | 0 (0.0) | 0.243 |
| Abdominal pain | 23 (22.8) | 23 (44.2) | 0 (0.0) | 0.000 |
| Nausea | 33 (32.7) | 28 (53.8) | 5 (10.2) | 0.000 |
| Vomiting | 26 (25.7) | 25 (48.1) | 1 (2.0) | 0.000 |
| Laboratory values on admission | | | | |
| WBC (x 10³/µl) n=98 | 8.5±0.2 | 9.2±0.2 | 7.7±0.2 | 0.008 |
| Platelets (x 10³/µl) n=98 | 236±96 | 246±165 | 225±147 | 0.092 |
| Creatinine (mg/dL) n=96 | 0.97±0.00 | 1.00±0.00 | 0.95±0.00 | 0.056 |
| INR n=97 | 1.14±0.01 | 1.07±0.00 | 1.21±0.02 | 0.672 |
| Fibrinogen (mg/dL) n=35 | 259±177 | 257±473 | 263±551 | 0.268 |
| Admission times (minutes) – median [IQR] | | | | |
| Snakebite to hospital | 45 [30-61] | 47 [30-78] | 43 [33-60] | 0.703 |
| Snakebite to ICU admission | 183 [127-260] | 169 [108-278] | 183 [135-209] | 0.939 |
| Hospital to ICU admission | 121 [81-179] | 133 [75-197] | 110 [90-169] | 0.793 |
| Snakebite to antivenin | | 119 [60-237] | | |
| Hospital admission to antivenin | | 48 [24-222] | | |
| Snakebite to last antivenin | | 195 [110-324] | | |

CI: confidence interval; ICU: intensive care unit; INR: international normalized ratio; IQR: interquartile range; WBC: white blood cells.

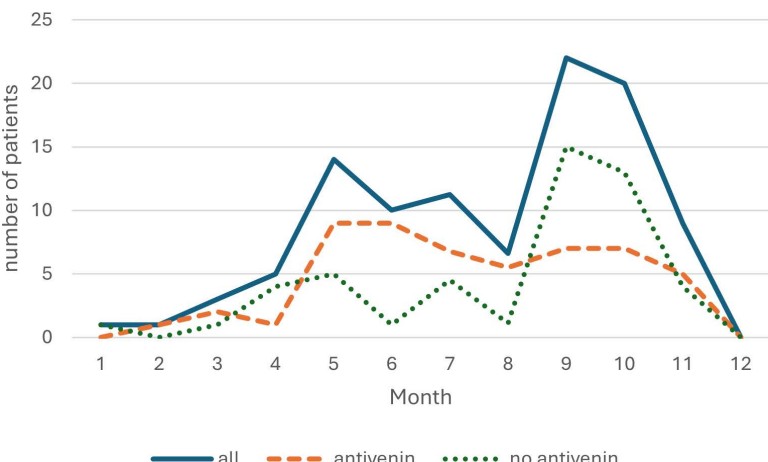

**Fig 3. Seasonal envenomation variation.** Note: due to missing data from July-August 2023 this data has been interpolated.

**Table 2. Treatment and course of hospitalization.**

| Variable | All patients n = 101 | Antivenin n = 52 | No antivenin n = 49 | p-value |
|---|---|---|---|---|
| Total antivenin dose (ml) – median [IQR] | | 15 [10-22.5] | | |
| Total antivenin treatments – median [IQR] | | 1 [1–2] | | |
| Other treatments received – n (%) | | | | |
| Vasoactive medication | 7 (6.9) | 7 (13.5) | 0 (0.0) | 0.013 |
| Supplemental oxygen | 5 (5.0) | 5 (9.6) | 0 (0.0) | 0.057 |
| Invasive ventilation | 2 (2.0) | 2 (3.8) | 0 (0.0) | 0.495 |
| Blood products | 2 (2.0) | 2 (3.8) | 0 (0.0) | 0.495 |
| Antiemetics | 19 (18.8) | 15 (28.8) | 4 (8.2) | 0.010 |
| Non-opiate analgesia | 50 (49.5) | 35 (67.3) | 15 (30.6) | 0.001 |
| Opiate analgesia | 41 (40.6) | 28 (53.8) | 13 (26.5) | 0.008 |
| Surgical intervention | 4 (4.0) | 3 (5.8) | 1 (2.0) | 0.618 |
| Laboratory values – median [IQR] | | | | |
| WBC max (x $10^3$/µl) | 9.6 [7.9-11.9] | 11.1 [9.2-13.4] | 8.4 [7-10.1] | 0.001 |
| Platelets nadir (x $10^3$/µl) | 175 [133-213] | 151 [118-188] | 186 [161-240] | 0.002 |
| Creatinine max (mg/dl) | 0.96 [0.85-1.1] | 0.99 [0.86-1.12] | 0.95 [0.85-1.05] | 0.091 |
| INR max | 1.06 [1.02-1.15] | 1.09 [1.04-1.18] | 1.03 [1.01-1.11] | 0.566 |
| Fibrinogen nadir (mg/dl) | 241 [206-289] | 235 [201-293] | 252 [231-285] | 0.414 |
| Outcomes – median [IQR] | | | | |
| ICU LOS | 2 [2–3] | 3 [2–3] | 2 [1–2] | 0.001 |
| Hospital LOS | 3 [2–4] | 4 [3–5] | 3 [2–4] | 0.001 |

ICU: intensive care unit; INR: international normalized ratio; IQR: interquartile range; WBC: white blood cells.

platelets was noted in most patients over the course of the hospitalization, with 49/52 (94.2%) patients in the antivenin group and 33/49 (67.3%) of patients in the non-antivenin group developing a platelet reduction relative to admission (p = 0.001), with a median nadir of 151 [118–188] x $10^3$ per µl in the antivenin versus 186 [161–240] x $10^3$ per µl in the non-antivenin group, p < 0.001. Three patients in the antivenin group developed a severe thrombocytopenia of less than

50 x 10$^3$ per µl, with no cases of severe thrombocytopenia in the non-antivenin group. No patients developed major bleeding or required platelet transfusion.

No substantial pathologies or differences in coagulation factors were noted between groups, with a median maximal INR of 1.06 [1.02–1.15] and a median fibrinogen nadir of 241 [206–289] mg/dl for the cohort. One patient presented with an INR of 10.4, which was thought to be due to laboratory error and, in fact, normalized without treatment upon repeat collection. One patient on chronic warfarin anticoagulation presented with an INR of 3.49 and another patient developed an INR of 1.88 during course of hospitalization which normalized without treatment. None of these patients received antivenin or blood products. No patient developed clinically significant fibrinogen abnormalities.

Median maximal creatinine was 0.96 [0.85–1.1] with no differences between groups. No patient developed acute kidney injury during their hospitalization.

Median length of stay in the ICU was three [2–3] days for the antivenin and two [1–2] days for the non-antivenin group (p<0.001). Median hospital length of stay was four [3–5] days for the antivenin group and three [2–4] days for the non-antivenin group (Table 2). No patients died during the course of their hospitalization.

## Discussion

This study of 101 patients diagnosed with *D. palaestinae* envenomation is the largest published cohort of human *D. palaestinae* envenomations to date. It shows that a dose-scaling antivenin protocol can be safely used in the treatment of envenomation, thus minimizing side effects and overall morbidity compared to the currently accepted fixed-dose protocol [5], reducing costs and reducing antivenin use, especially in periods of scarcity [11].

In the current study, local symptoms were treated conservatively, without use of antivenin [6,7,9] and treatment of both mild and severe systemic symptoms employed dose scaling antivenin, with an initial 10 ml dose of antivenin and repeat dosing in patients with ongoing systemic symptoms, in accordance with treatment response. In our study, 52 of 101 (51.5%) patients received antivenin, with 26 of those receiving 10 ml of antivenin and another 13 receiving 20 ml. This is in contrast with the standard protocol, in which most or all of these patients would have received 50 ml of antivenin [5,6,13]. This protocol resulted in an estimated cost saving of over $500,000, as well as an estimated two fewer allergic reactions, including one less anaphylaxis, with no increased morbidity or mortality related to the reduced dose protocol.

In the current study, nearly all patients diagnosed with *D. palaestinae* envenomation develop swelling and a majority suffering from site pain, with a higher percentage of pain reported among those requiring antivenin. We also report abdominal symptoms, respiratory distress, hypotension and with one case of reduced consciousness associated with envenomation. This is similar to previous studies [5,9–11]. Unlike those studies, our patients did not develop convulsions, coagulopathy or necrosis, possibly due to early antivenin treatment when necessary, with a median time between snakebite and initial antivenin treatment of 119 [60–237] minutes. Similar to previous studies, a majority of our cohort showed a reduction in platelets over the course of the hospitalization [22]. This was true whether or not the patient was treated with antivenin (though patients who received antivenin had a more substantial platelet reduction), and, similar to some [23], but unlike other [24], reports in other Viperidae species, there was no apparent response to antivenin. This may be due to venom-induced macrophage and hepatocyte sequesterization and clearance of platelets, as shown in other species of Viperidae [25,26].

Similar to previous studies [10], we show that some patients show a latent development of symptoms and repeat symptom occurrence. Thus, the maximum time between snakebite and first antivenin administration was 17.8 hours and the maximum time until last antivenin administration (due to repeat symptoms) was 64.3 hours.

This study has several limitations. Firstly, it is a retrospective data analysis of an unmatched cohort, and, therefore, its conclusions should be approached with caution. Furthermore, though, in our institution, *D. palaestinae* is the most common cause of snakebite and practically the only cause of snakebite envenomation, species identification was made by the treating clinician and an expert in snake taxonomy was not available for species confirmation. This may have lead

to species misidentification. Finally, though a comparison has been made between those who did and did not receive antivenin, due to the inherent clinical differences between these groups, no true control group was employed. Rather, in order to evaluate the antivenin protocol of the current study, historical controls from the scientific literature were employed, which also has substantial limitations. Nevertheless, we demonstrated no morbidity or mortality related to the protocol.

## Conclusions

A dose-scaling antivenin protocol can be safely used in the treatment of *D. palaestinae* envenomation, thus reducing morbidity related to antivenin administration as well as reducing costs. This suggests the need for a randomized control trial comparing a fixed dose regimen to the escalation protocol described in this study and suggests the development of similar sliding scale protocols for envenomations due to other snake species.

## Author contributions

**Conceptualization:** Daniel J. Jakobson, Frederic S Zimmerman.

**Data curation:** Zurab Zakariashvili, Enzo F. Galicia H., Mohammad Abu Issa, Frederic S Zimmerman.

**Formal analysis:** Daniel J. Jakobson, Mohammad Abu Issa, Frederic S Zimmerman.

**Investigation:** Daniel J. Jakobson, Zurab Zakariashvili, Enzo F. Galicia H., Mohammad Abu Issa, Miguel M. Glatstein, Frederic S Zimmerman.

**Methodology:** Daniel J. Jakobson, Frederic S Zimmerman.

**Supervision:** Daniel J. Jakobson, Miguel M. Glatstein, Frederic S Zimmerman.

**Writing – original draft:** Daniel J. Jakobson, Frederic S Zimmerman.

**Writing – review & editing:** Daniel J. Jakobson, Miguel M. Glatstein, Frederic S Zimmerman.

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
