## [Decision Letter · Decision Letter 0]

PONE-D-25-04569A dose scaling antivenin protocol in treatment of Daboia palaestinae envenomation may reduce morbidity and costs.PLOS ONE

Dear Dr. Zimmerman,

Thank you for submitting your manuscript to PLOS ONE. After careful consideration, we feel that it has merit but does not fully meet PLOS ONE’s publication criteria as it currently stands. Therefore, we invite you to submit a revised version of the manuscript that addresses the points raised during the review process.

We look forward to receiving your revised manuscript.

Kind regards,

Timothy Omara

Academic Editor

PLOS ONE

Journal Requirements:

2. Please include captions for your Supporting Information files at the end of your manuscript, and update any in-text citations to match accordingly. Please see our Supporting Information guidelines for more information: http://journals.plos.org/plosone/s/supporting-information .

Additional Editor Comments:

Dear authors,

Reviewers have advised that your manuscript could be reconsidered after substantial revision. May I also ask you to:

Review the suggestions I have made in the attached manuscript file.Revise the following statements in the INTRODUCTION:Distribution of venomous snakes: To my knowledge, venomous snakes are indeed found on all continents except Antarctica, as the extreme cold environment there is unsuitable for reptiles. However, some island countries (Iceland, Ireland, Greenland, and New Zealand) are other likely exceptions; they naturally lack native (and venomous) snake species.Clinical signs, symptoms and symptom severity of envenomation: This depends on the composition of the venom, which is in turn dependent on the species (snake’s gender, age, prey availability, diet, geographic location, among other factors).

3. Ask a proficient colleague or use an English language editing service to check the draft for grammatical fixes and English language 

Reviewers' comments:

Reviewer's Responses to Questions

**Comments to the Author**

1. Is the manuscript technically sound, and do the data support the conclusions?

Reviewer #1: Partly

Reviewer #2: Yes

2. Has the statistical analysis been performed appropriately and rigorously? 

Reviewer #1: No

Reviewer #2: Yes

3. Have the authors made all data underlying the findings in their manuscript fully available?

Reviewer #1: No

Reviewer #2: No

4. Is the manuscript presented in an intelligible fashion and written in standard English?

Reviewer #1: Yes

Reviewer #2: Yes

5. Review Comments to the Author

Reviewer #1: Review Comments to the Author

General

This article addresses the knowledge gap on D palaestinae bite treatment in Mediterranean regions. This article provides suggestions strategies for improvement in snakebite control in Israel, which can be translated into other countries in the east Mediterranean regions where this snake species inhabits and envenoms humans often. Hence, this paper can be a reference for snakebite management in the eastern Mediterranean countries where snakebite problem due to the species discussed in this article is noticeable. However, authors need to revise this manuscript particularly introduction, methods, and discussion sections extensively.

Overall, this manuscript is well written. However, the English use needs to improve in several occasions. Please, use continuous line number to make peer-review and your revisions followed by reviewers' comment easy. There should be a single space between words [e.g., toDaboia would be to Daboia in the fifth reference: 5. Wang AH, Gordon D, Drescher MJ, Shiber S. Is there a benefit for administration of antivenom following local and mild systemic reactions toDaboia (Vipera) Palaestinaesnake bites? Toxin Reviews. 2018;39(1):52-6. doi: 10.1080/15569543.2018.1477162.]

Please, italicize all genus and species names (initial letter of species name is never a capital letter), in the main body of the manuscript as well as in the reference section. For example: see: 5. Wang AH, Gordon D, Drescher MJ, Shiber S. Is there a benefit for administration of antivenom following local and mild systemic reactions toDaboia (Vipera) Palaestinaesnake bites? Toxin Reviews. 2018;39(1):52-6. doi: 10.1080/15569543.2018.1477162.

Please, spell all numbers below 10 [e.g., 2 for two] in Abstract and throughout the mainbody of the manuscript.

Specific

Abstract

Please, replace " Results

101 patients were included. 45 minutes [median; interquartile range: 30-61 minutes]" with " Results

101 patients were included. A median of 45 minutes [interquartile range: 30-61 minutes]".

Introduction

The first two lines of the first paragraph need clarification and use of proper reference.

In "Snake envenomation is a widespread cause of mortality and morbidity worldwide, with snake bites resulting in 50 – 130,000 deaths annually", where these deaths occurred? Is it incidence in the eastern Mediterranean countries where the species focused in this study envenom people commonly? Globally, it is reported that severe envenoming results in between 81,000 and 138,000 deaths annually. Please, follow up " Kasturiratne A, Wickremasinghe AR, de Silva N, Gunawardena NK, Pathmeswaran A, Premaratna R, Savioli L, Lalloo DG, de Silva HJ: The global burden of snakebite: a literature analysis and modelling based on regional estimates of envenoming and deaths. PLoS Med 2008, 5(11):e218." to know about the most comprehensive and recent global or regional impact of snakebite.

Please, replace "poisonous" with "venomous" in the second paragraph and elsewhere in this manuscript it is because poison and venom are different terminology.

Please, merge the fourth and the fifth paragraph. Also, the 11th reference did not focus on shortage of antivenom. So, please, improve the section "[4, 5, 11]. Furthermore, certain

antivenins have historically been in short supply [11]. Moreover," with the precise referencing. Isn't this study "11. Tirosh-Levy S, Solomovich-Manor R, Comte J, Nissan I, Sutton GA, Gabay A, et al.

Daboia (Vipera) palaestinae Envenomation in 123 Horses: Treatment and Efficacy of

Antivenom Administration. Toxins (Basel). 2019;11(3). Epub 20190319. doi:

10.3390/toxins11030168. PubMed PMID: 30893807; PubMed Central PMCID:

PMCPMC6468471." focused to effectiveness of antivenom use in horses? So, referencing use looks irrelevant herein. Please, reconfirm and improve it accordingly.

Please, merge the second last and the last paragraph.

Methods

The method section needs improvement further. Create a subsection for study sites and vulnerable human population in the service area of study hospital. The distribution of snakebite cases across the service area of this hospital should be linked with the climatic conditions, topography, and the variety of medically relevant snake species inhabiting these areas. It is better to illustrate these areas in Figure. You aimed to illustrate antivenom dosing which depend on venom and antivenom quality and geographical variations of snake species distribution and the use of venom from snake inhabiting a certail locality. To discuss on this issue, you need to provide its temperature, altitude, and the dominant snake species (is D plaestinae a dominant one in Israel?).

Further, please, briefly mention about the "identification of snake specimens and their vouchers" in this cohort study. Were those snakes identified by expert in snake taxonomy? Were these a list of non-expert identified snake, too? Please, mention about the identification issues as there are several report of misidentification of snake species by medical professionals in other nations. e.g., Namal Rathnayaka, R. M. M. K., Nishanthi Ranathunga, P. E. A., and Kularatne, S. a. M. (2021). Paediatric cases of Ceylon Krait (Bungarus ceylonicus) bites and some similar looking non-venomous snakebites in Sri Lanka: misidentification and antivenom administration. Toxicon, 198: 143–150.

The second last paragraph would be subsection: Data analysis. This section needs re-writing after the normality tests of data and ensuring the representativeness of samples (SBE cases by this particular species). Median would be better descriptive statistics if data were skewed than the means to describe the study population. Please, list the categorical variables used in the Chi square test or Fisher's exact test and Students T-test or Mann-Whitney U tests in the parentheses, respectively. This increases the repeatability of this study.

Results

"During the study period, 114 patients were admitted to Barzilai University Medical Center

with a snake bite diagnosis. After chart review, 10 patients were eliminated after ruling out

D. palaestinae envenomation, 2 more patients were eliminated due to initial treatment in

another hospital and 1 minor patient was eliminated. Thus," goes to method section under the subsection: Inclusion exclusion criteria. This is the place to present only the cases included in this study.

Discussion

This section needs re-writing. Please, improve the first paragraph " The primary and most effective treatment of snake bite envenomation, including that

resulting from D. palaestinae bite, continues to be administration of specific or polyvalent

snake antivenin, which drastically reduces mortality and morbidity [8, 9]. However, antivenin

administration is costly, with a single fixed dose regimen estimated cost of $7000, with

repeat administration sometimes required [4, 5, 11]. Furthermore, certain antivenins have

historically been in short supply [11]. These limitations can form substantial barriers to

treatment access, especially in limited-resource settings. Moreover, antivenin administration

can itself be associated with substantial morbidity, including delayed reactions up to a

month after administration [4, 8, 10]. Thus, it is important to define the minimal necessary

dose for effective reversal of envenomation, as well as to define patient populations who can be safely treated without antivenin administration. ".

It illustrates tremendously high costs which is highly contrasting to similar cost of treating envenomation in Nepal (Pandey, D. P., Adhikari, B., Pandey, P., Sapkota, K., Bhusal, M., Kandel, P., Shrestha, D. L., and Shrestha, B. R. (2024). Cost of snakebite and its impact on household economy in southern Nepal. Am J Trop Med Hyg, 10.4269/ajtmh.24-0399.). The first paragraph should discuss on the core findings of this study. Accordingly, you need to attract your readers with similar or contrasting findings and your insight regarding this uniqueness.

Conclusions

Please, re-write the conclusion as:

A dose-scaling antivenin protocol can be safely used in the treatment of D. palaestinae envenomation, thus reducing the treatment costs and morbidity of envenomed patients. This suggests the need for a randomized control trial comparing a fixed dose regimen to the escalation

protocol as described in this study and for developing similar sliding scale protocols for envenomations due to other snake species, too.

Acknowledgement:

Did you miss this section?

Please, check and mention it if you find it relevant.

Reviewer #2: Thank you for the opportunity to review this manuscript. The authors present a retrospective study of over 100 patients with D. palaestinae envenomation and conclude that the sliding scale antivenom administration was effective and safe. Although this study does not directly compare with the classic protocol or patients without antivenom for systemic signs of envenomation (as acknowledged in the limitations), the findings appear helpful for considering future antivenom treatment strategies. However, I have several concerns with the current version of the manuscript.

Major Concerns:

1. Please clarify if "time between envenomation and hospital" refers to time between bite and hospital visit. Did the authors determine the time of onset of envenomation signs?

2. The classic protocol for antivenom with a fixed dose of 50ml - how was this established? The authors should describe the rationale.

3. Please provide more details about the antivenom: was it monovalent or polyvalent? What antivenoms are available in the region, who are the producers, and which specific antivenom was used in this study?

4. Although the proposed protocol is described in the main text, a figure with an algorithm might help facilitate understanding.

5. How many patients were definitively identified as having D. palaestinae envenomation, and by what means were they identified?

6. For patients with adverse reactions, what amount of antivenom was given before the reaction occurred? Was there any history of equine serum exposure or allergic reactions previously?

7. While no difference in mortality between groups was noted, were there any after-effects of local envenomation such as deformity, functional impairment, or chronic pain?

Minor Concerns:

1. In the Introduction, more information about the geographical distribution of prevalent snakes other than D. palaestinae would be helpful to understand the regional situation.

2. In the study hospital, did all patients stay in the ICU after being hospitalized? What were the discharge criteria?

3. Regarding bite site location, is "Hand/Leg" the correct classification? Does "Hand" include forearm/upper arm, and does "Leg" include foot or toe?

4. Some sentences in the Methods section are repetitive and should be consolidated.

5. The first paragraph of the Discussion is redundant and should be omitted or relocated to the Introduction.

6. Are there any previous reports of sliding scale protocols for snakebite antivenom elsewhere?

6. PLOS authors have the option to publish the peer review history of their article (what does this mean? ). If published, this will include your full peer review and any attached files.

**Do you want your identity to be public for this peer review?** For information about this choice, including consent withdrawal, please see our Privacy Policy .

Reviewer #1: **Yes: ** Deb Prasad Pandey

Reviewer #2: No

---

## [Author Response · Author response to Decision Letter 1]

20 Apr 2025

- the paper has been updated in accordance with PLOS ONE’s style requirements

- No Supporting Information files have been included in the paper

Additional Editor Comments:

Dear authors,

Reviewers have advised that your manuscript could be reconsidered after substantial revision. May I also ask you to:

1. Review the suggestions I have made in the attached manuscript file.

- The suggestions have been reviewed, and the manuscript has been updated accordingly. We accept changes made via track changes.

2. Revise the following statements in the INTRODUCTION:

3. Distribution of venomous snakes: To my knowledge, venomous snakes are indeed found on all continents except Antarctica, as the extreme cold environment there is unsuitable for reptiles. However, some island countries (Iceland, Ireland, Greenland, and New Zealand) are other likely exceptions; they naturally lack native (and venomous) snake species.

- The paragraph has been updated accordingly

4. Clinical signs, symptoms and symptom severity of envenomation: This depends on the composition of the venom, which is in turn dependent on the species (snake’s gender, age, prey availability, diet, geographic location, among other factors).

- The paragraph has been updated accordingly

3. Ask a proficient colleague or use an English language editing service to check the draft for grammatical fixes and English language

- the draft has been reviewed and corrected by a proficient English speaker.

Reviewers' comments:

Reviewer's Responses to Questions

Comments to the Author

1. Is the manuscript technically sound, and do the data support the conclusions?

Reviewer #1: Partly

Reviewer #2: Yes

- Thank you for feedback. The paper has been revised according to reviewer comments.

2. Has the statistical analysis been performed appropriately and rigorously?

Reviewer #1: No

Reviewer #2: Yes

- Thank you for feedback. The statistical analysis section has been revised in accordance with reviewer comments.

3. Have the authors made all data underlying the findings in their manuscript fully available?

Reviewer #1: No

Reviewer #2: No

-As noted in the original submission, all data files are available from the Open Science Foundation database (accession number osf.io/2p685).

4. Is the manuscript presented in an intelligible fashion and written in standard English?

Reviewer #1: Yes

Reviewer #2: Yes

- Thank you.

5. Review Comments to the Author

Reviewer #1: Review Comments to the Author

General

This article addresses the knowledge gap on D palaestinae bite treatment in Mediterranean regions. This article provides suggestions strategies for improvement in snakebite control in Israel, which can be translated into other countries in the east Mediterranean regions where this snake species inhabits and envenoms humans often. Hence, this paper can be a reference for snakebite management in the eastern Mediterranean countries where snakebite problem due to the species discussed in this article is noticeable. However, authors need to revise this manuscript particularly introduction, methods, and discussion sections extensively.

Overall, this manuscript is well written. However, the English use needs to improve in several occasions. Please, use continuous line number to make peer-review and your revisions followed by reviewers' comment easy. There should be a single space between words [e.g., toDaboia would be to Daboia in the fifth reference: 5. Wang AH, Gordon D, Drescher MJ, Shiber S. Is there a benefit for administration of antivenom following local and mild systemic reactions toDaboia (Vipera) Palaestinaesnake bites? Toxin Reviews. 2018;39(1):52-6. doi: 10.1080/15569543.2018.1477162.]

- This has been corrected

Please, italicize all genus and species names (initial letter of species name is never a capital letter), in the main body of the manuscript as well as in the reference section. For example: see: 5. Wang AH, Gordon D, Drescher MJ, Shiber S. Is there a benefit for administration of antivenom following local and mild systemic reactions toDaboia (Vipera) Palaestinaesnake bites? Toxin Reviews. 2018;39(1):52-6. doi: 10.1080/15569543.2018.1477162.

- This has been corrected

Please, spell all numbers below 10 [e.g., 2 for two] in Abstract and throughout the mainbody of the manuscript.

- This has been changed

Specific

Abstract

Please, replace " Results

101 patients were included. 45 minutes [median; interquartile range: 30-61 minutes]" with " Results

101 patients were included. A median of 45 minutes [interquartile range: 30-61 minutes]".

- This has been changed

Introduction

The first two lines of the first paragraph need clarification and use of proper reference.

In "Snake envenomation is a widespread cause of mortality and morbidity worldwide, with snake bites resulting in 50 – 130,000 deaths annually", where these deaths occurred? Is it incidence in the eastern Mediterranean countries where the species focused in this study envenom people commonly? Globally, it is reported that severe envenoming results in between 81,000 and 138,000 deaths annually. Please, follow up " Kasturiratne A, Wickremasinghe AR, de Silva N, Gunawardena NK, Pathmeswaran A, Premaratna R, Savioli L, Lalloo DG, de Silva HJ: The global burden of snakebite: a literature analysis and modelling based on regional estimates of envenoming and deaths. PLoS Med 2008, 5(11):e218." to know about the most comprehensive and recent global or regional impact of snakebite.

-The paragraph has been clarified and new references reviewed and added.

Please, replace "poisonous" with "venomous" in the second paragraph and elsewhere in this manuscript it is because poison and venom are different terminology.

-This has been corrected

Please, merge the fourth and the fifth paragraph. Also, the 11th reference did not focus on shortage of antivenom. So, please, improve the section "[4, 5, 11]. Furthermore, certain

antivenins have historically been in short supply [11]. Moreover," with the precise referencing. Isn't this study "11. Tirosh-Levy S, Solomovich-Manor R, Comte J, Nissan I, Sutton GA, Gabay A, et al.

Daboia (Vipera) palaestinae Envenomation in 123 Horses: Treatment and Efficacy of

Antivenom Administration. Toxins (Basel). 2019;11(3). Epub 20190319. doi:

10.3390/toxins11030168. PubMed PMID: 30893807; PubMed Central PMCID:

PMCPMC6468471." focused to effectiveness of antivenom use in horses? So, referencing use looks irrelevant herein. Please, reconfirm and improve it accordingly.

The paragraphs have been merged and the references corrected

Please, merge the second last and the last paragraph.

- Paragraphs have been merged

Methods

The method section needs improvement further. Create a subsection for study sites and vulnerable human population in the service area of study hospital. The distribution of snakebite cases across the service area of this hospital should be linked with the climatic conditions, topography, and the variety of medically relevant snake species inhabiting these areas. It is better to illustrate these areas in Figure. You aimed to illustrate antivenom dosing which depend on venom and antivenom quality and geographical variations of snake species distribution and the use of venom from snake inhabiting a certail locality. To discuss on this issue, you need to provide its temperature, altitude, and the dominant snake species (is D plaestinae a dominant one in Israel?).

Further, please, briefly mention about the "identification of snake specimens and their vouchers" in this cohort study. Were those snakes identified by expert in snake taxonomy? Were these a list of non-expert identified snake, too? Please, mention about the identification issues as there are several report of misidentification of snake species by medical professionals in other nations. e.g., Namal Rathnayaka, R. M. M. K., Nishanthi Ranathunga, P. E. A., and Kularatne, S. a. M. (2021). Paediatric cases of Ceylon Krait (Bungarus ceylonicus) bites and some similar looking non-venomous snakebites in Sri Lanka: misidentification and antivenom administration. Toxicon, 198: 143–150.

- Issues of climatic conditions, topography, relevant snake species and snake identification have now been addressed, and a relevant figure has been added.

The second last paragraph would be subsection: Data analysis. This section needs re-writing after the normality tests of data and ensuring the representativeness of samples (SBE cases by this particular species). Median would be better descriptive statistics if data were skewed than the means to describe the study population. Please, list the categorical variables used in the Chi square test or Fisher's exact test and Students T-test or Mann-Whitney U tests in the parentheses, respectively. This increases the repeatability of this study.

- The relevant paragraph has been labeled as subsection: Data analysis.

- As further clarified in Methods, Daboia palaestinae represents the only relevant SBE in the environs of our institution. Since all SBEs admitted to our institution during the study period were included in the current study, our sample represents the entire incidence of SBE in our region.

- Statistical tests used for each variable have now been clarified.

- Medians are used to describe skewed data (admission times) whereas means are used to describe normally distributed data (age, laboratory values)

Results

"During the study period, 114 patients were admitted to Barzilai University Medical Center

with a snake bite diagnosis. After chart review, 10 patients were eliminated after ruling out

D. palaestinae envenomation, 2 more patients were eliminated due to initial treatment in

another hospital and 1 minor patient was eliminated. Thus," goes to method section under the subsection: Inclusion exclusion criteria. This is the place to present only the cases included in this study.

- This has been moved to the Methods section under the relevant subsection.

Discussion

This section needs re-writing. Please, improve the first paragraph " The primary and most effective treatment of snake bite envenomation, including that

resulting from D. palaestinae bite, continues to be administration of specific or polyvalent

snake antivenin, which drastically reduces mortality and morbidity [8, 9]. However, antivenin

administration is costly, with a single fixed dose regimen estimated cost of $7000, with

repeat administration sometimes required [4, 5, 11]. Furthermore, certain antivenins have

historically been in short supply [11]. These limitations can form substantial barriers to

treatment access, especially in limited-resource settings. Moreover, antivenin administration

can itself be associated with substantial morbidity, including delayed reactions up to a

month after administration [4, 8, 10]. Thus, it is important to define the minimal necessary

dose for effective reversal of envenomation, as well as to define patient populations who can be safely treated without antivenin administration. ".

It illustrates tremendously high costs which is highly contrasting to similar cost of treating envenomation in Nepal (Pandey, D. P., Adhikari, B., Pandey, P., Sapkota, K., Bhusal, M., Kandel, P., Shrestha, D. L., and Shrestha, B. R. (2024). Cost of snakebite and its impact on household economy in southern Nepal. Am J Trop Med Hyg, 10.4269/ajtmh.24-0399.). The first paragraph should discuss on the core findings of this study. Accordingly, you need to attract your readers with similar or contrasting findings and your insight regarding this uniqueness.

- The Discussion section has been changed as suggested and reference to costs in other locales has been added.

Conclusions

Please, re-write the conclusion as:

A dose-scaling antivenin protocol can be safely used in the treatment of D. palaestinae envenomation, thus reducing the treatment costs and morbidity of envenomed patients. This suggests the need for a randomized control trial comparing a fixed dose regimen to the escalation

protocol as described in this study and for developing similar sliding scale protocols for envenomations due to other snake species, too.

- This has been changed as suggested.

Acknowledgement:

Did you miss this section?

Please, check and mention it if you find it relevant.

- The acknowledgements section is not relevant for the current paper

Reviewer #2: Thank you for the opportunity to review this manuscript. The authors present a retrospective study of over 100 patients with D. palaestinae envenomation and conclude that the sliding scale antivenom administration was effective and safe. Although this study does not directly compare with the classic protocol or patients without antivenom for systemic signs of envenomation (as acknowledged in the limitations), the findings appear helpful for considering future antivenom treatment strategies. However, I have several concerns with the current version of the manuscript.

Major Concerns:

1. Please clarify if "time between envenomation and hospital" refers to time between bite and hospital visit. Did the authors determine the time of onset of envenomation signs?

-The time refers to time between bite and hospital visit. This has now been clarified. Time of onset of envenomation signs was not noted.

2. The classic protocol for antivenom with a fixed dose of 50ml - how was this established? The authors should describe the rationale.

-This has now been described and references.

3. Please provide more details about the antivenom: was it monovalent or polyvalent? What antivenoms are available in the region, who are the producers, and which specific antivenom was used in this study?

-This has now been provided

4. Although the proposed protocol is described in the main text, a figure with an algorithm might help facilitate understanding.

-This has been presented in Figure 2

5. How many patients were definitively identified as having D. palaestinae envenomation, and by what means were they identified?

-Unf

---

## [Decision Letter · Decision Letter 1]

PONE-D-25-04569R1A dose scaling antivenin protocol in treatment of Daboia palaestinae envenomation may reduce morbidity and costs.PLOS ONE

Dear Dr. Zimmerman,

Thank you for submitting your manuscript to PLOS ONE. After careful consideration, we feel that it has merit but does not fully meet PLOS ONE’s publication criteria as it currently stands. Therefore, we invite you to submit a revised version of the manuscript that addresses the points raised during the review process.

We look forward to receiving your revised manuscript.

Kind regards,

Timothy Omara

Academic Editor

PLOS ONE

Journal Requirements:

Reviewers' comments:

Reviewer's Responses to Questions

**Comments to the Author**

1. If the authors have adequately addressed your comments raised in a previous round of review and you feel that this manuscript is now acceptable for publication, you may indicate that here to bypass the “Comments to the Author” section, enter your conflict of interest statement in the “Confidential to Editor” section, and submit your "Accept" recommendation.

Reviewer #1: (No Response)

2. Is the manuscript technically sound, and do the data support the conclusions?

Reviewer #1: Yes

3. Has the statistical analysis been performed appropriately and rigorously? 

Reviewer #1: No

4. Have the authors made all data underlying the findings in their manuscript fully available?

Reviewer #1: No

5. Is the manuscript presented in an intelligible fashion and written in standard English?

Reviewer #1: Yes

6. Review Comments to the Author

Reviewer #1: Review Comments to the Authors

General

Authors have improved the manuscript noticeably. However, method section needs further improvement. I would suggest authors for minor revise of introduction and method section.

Overall, this manuscript is well written. I would suggest authors to spell all numbers below 10 [e.g., 2 for two].

I could not track the given database in author response section. Please, give the url to enable accessing these data files.

Specific

Introduction

Minor improvement is needed in this section. Please, replace

"... with 100-300 snakebites reported in Israel annually [6, 7]"

with

"... with 100-300 envenomations due to bites by this species from Israel annually [6, 7]".

Based on citation of 6th and 7th references, the 100 to 300 cases would be envenomed cases not snakebites in general. If this figure represent both venomous and nonvenomous cases, please, re-write this section more clearly.

Methods

The method section still needs improvement.

Authors have mentioned to address about the confirmation of snake species involved in bite in response to the reviewer section. But, they have not implemented it in the main body of the draft. Did they conform 101 cases by using symptomatic diagnosis or confirmed with involved snake species or snake venom detection kit or others? So, please, mention about the "identification of snake specimens and their vouchers" in this study. Were those snakes identified by expert in snake taxonomy? Were these a list of non-expert identified snake, too? Please, mention about the identification issues as there are several report of misidentification of snake species by medical professionals.

The data analysis (page 10-11) section still needs improvement. Authors look that they did not use the normality tests which orient authors to conform parametric or nonparametric tests. Additionally, they have not addressed the logics needed to use inferential statistics without explaining about the sampling strategies, sample size, and representativeness of populations by the selected cases envenomed by this particular species.

Authors compared proportions of certain variables using the χ2-score or the Fisher’s exact test. Please, list the categorical variables that you used in the Chi square test and Fisher's exact test distinctly. This helps to follow up the results that you present in another section.

Also, they compared continuous variables using Student’s t-test or the Mann-Whitney-Wilcoxon test. If statistical assumption to use these inferential tests meet with your data set, please, list the particular variables used in the Students T-test and the Mann-Whitney U tests in the parentheses.

To increases the repeatability of this study, improvement of method is still necessary.

7. PLOS authors have the option to publish the peer review history of their article (what does this mean? ). If published, this will include your full peer review and any attached files.

**Do you want your identity to be public for this peer review?** For information about this choice, including consent withdrawal, please see our Privacy Policy .

Reviewer #1: **Yes: ** Deb Prasad Pandey

---

## [Author Response · Author response to Decision Letter 2]

26 May 2025

Journal Requirements:

- No retracted papers have been cited

Reviewers' comments:

Reviewer's Responses to Questions

Comments to the Author

3. Has the statistical analysis been performed appropriately and rigorously?

Reviewer #1: No

- The statistical analysis has been updated as per comment below

4. Have the authors made all data underlying the findings in their manuscript fully available?

Reviewer #1: No

- All anonymized data is fully available without restriction at https://osf.io/2p685/

6. Review Comments to the Author

Reviewer #1: Review Comments to the Authors

General

Authors have improved the manuscript noticeably. However, method section needs further improvement. I would suggest authors for minor revise of introduction and method section.

Overall, this manuscript is well written. I would suggest authors to spell all numbers below 10 [e.g., 2 for two].

-All numbers below 10 have been spelled

I could not track the given database in author response section. Please, give the url to enable accessing these data files.

- All anonymized data is fully available without restriction at https://osf.io/2p685/

Specific

Introduction

Minor improvement is needed in this section. Please, replace

"... with 100-300 snakebites reported in Israel annually [6, 7]"

with

"... with 100-300 envenomations due to bites by this species from Israel annually [6, 7]".

Based on citation of 6th and 7th references, the 100 to 300 cases would be envenomed cases not snakebites in general. If this figure represent both venomous and nonvenomous cases, please, re-write this section more clearly.

-The sentence has been modified as per suggestion

Methods

The method section still needs improvement.

Authors have mentioned to address about the confirmation of snake species involved in bite in response to the reviewer section. But, they have not implemented it in the main body of the draft. Did they conform 101 cases by using symptomatic diagnosis or confirmed with involved snake species or snake venom detection kit or others? So, please, mention about the "identification of snake specimens and their vouchers" in this study. Were those snakes identified by expert in snake taxonomy? Were these a list of non-expert identified snake, too? Please, mention about the identification issues as there are several report of misidentification of snake species by medical professionals.

- Diagnosis of D. palaestinae envenomation was made by the treating clinician by clinical symptoms, species description and, when available, by identification of the species either directly or by photograph. Unfortunately, an expert in snake taxonomy was not available for species confirmation. Identification has now been discussed in methods and the limitations in identification have now been noted in the limitations section.

The data analysis (page 10-11) section still needs improvement. Authors look that they did not use the normality tests which orient authors to conform parametric or nonparametric tests. Additionally, they have not addressed the logics needed to use inferential statistics without explaining about the sampling strategies, sample size, and representativeness of populations by the selected cases envenomed by this particular species.

Authors compared proportions of certain variables using the χ2-score or the Fisher’s exact test. Please, list the categorical variables that you used in the Chi square test and Fisher's exact test distinctly. This helps to follow up the results that you present in another section.

Also, they compared continuous variables using Student’s t-test or the Mann-Whitney-Wilcoxon test. If statistical assumption to use these inferential tests meet with your data set, please, list the particular variables used in the Students T-test and the Mann-Whitney U tests in the parentheses.

To increase the repeatability of this study, improvement of method is still necessary.

- Thank you for your insightful comments regarding the statistical analysis. We have revised the Methods section accordingly to improve transparency and reproducibility:

1. We have now performed normality tests (Shapiro-Wilk) on continuous variables to determine the appropriate statistical tests. Variables that followed a normal distribution were analyzed using Student’s t-test, while non-normally distributed variables were analyzed using the Mann–Whitney U test.

2. We compared categorical variables using the Chi-square test, or Fisher’s exact test where expected cell counts were <5.

3. Specifically:

o Continuous variables were analyzed using:

Student’s t-test for: age, laboratory values

Mann–Whitney U test for: admission times, length of ICU stay and length of hospital stay

o Categorical variables analyzed using:

Chi-square: Demographics, location of bite, symptoms, supplemental treatments

Fisher’s exact: Antivenin vs no-antivenin in low-frequency months

4. We clarified our sampling strategy: all envenomation cases by D. palaestinae during the study period (2014–2022) were included. Thus, the dataset represents a consecutive case series, not a random sample.

5. An expert in statistical analysis has re-reviewed the manuscript for accuracy and reproducibility.

---

## [Decision Letter · Decision Letter 2]

PONE-D-25-04569R2A dose scaling antivenin protocol in treatment of Daboia palaestinae envenomation may reduce morbidity and costs.PLOS ONE

Dear Dr. Zimmerman,

Thank you for submitting your manuscript to PLOS ONE. After careful consideration, we feel that it has merit but does not fully meet PLOS ONE’s publication criteria as it currently stands. Therefore, we invite you to submit a revised version of the manuscript that addresses the points raised during the review process.

We look forward to receiving your revised manuscript.

Kind regards,

Timothy Omara

Academic Editor

PLOS ONE

Journal Requirements:

Reviewers' comments:

Reviewer's Responses to Questions

**Comments to the Author**

1. If the authors have adequately addressed your comments raised in a previous round of review and you feel that this manuscript is now acceptable for publication, you may indicate that here to bypass the “Comments to the Author” section, enter your conflict of interest statement in the “Confidential to Editor” section, and submit your "Accept" recommendation.

Reviewer #1: All comments have been addressed

2. Is the manuscript technically sound, and do the data support the conclusions?

Reviewer #1: Yes

3. Has the statistical analysis been performed appropriately and rigorously? 

Reviewer #1: Yes

4. Have the authors made all data underlying the findings in their manuscript fully available?

Reviewer #1: Yes

5. Is the manuscript presented in an intelligible fashion and written in standard English?

Reviewer #1: Yes

6. Review Comments to the Author

Reviewer #1: PONE-D-25-04569_R2_reviewer

Review Comments to the Author

General

Authors have adequately improved the manuscript. This manuscript can be accepted after minor revisions that I suggest authors below:

Specific

Methods

Were data collected from 2014-2023 OR from 2014-2022? Please, be consistent.

"five. . ": remove additional full stop herein.

7. PLOS authors have the option to publish the peer review history of their article (what does this mean? ). If published, this will include your full peer review and any attached files.

**Do you want your identity to be public for this peer review?** For information about this choice, including consent withdrawal, please see our Privacy Policy .

Reviewer #1: **Yes: ** Deb Prasad Pandey

---

## [Author Response · Author response to Decision Letter 3]

4 Jun 2025

Dear Editor,

We respectfully resubmit our manuscript: A dose scaling antivenin protocol in treatment of Daboia palaestinae envenomation may reduce morbidity and costs.

Below are our responses to requested changes in the manuscript. We hope the revised manuscript is found suitable for publication in PLOS ONE.

With thanks,

Frederic S. Zimmerman

Critical Care Unit

Shaare Zedek Medical Center

P.O. Box 3235

Jerusalem 91031, Israel

Tel +972-2-564-5922

Email: fzimmer@szmc.org.il

Daniel J. Jakobson

Director, Department of Intensive Care

Barzilai University Medical Center

2 Hahistadrout St

Ashkelon 7830604, Israel

Tel +972-3-674-3322

Email: danielj@bmc.gov.il

Journal Requirements:

- No retracted papers have been cited

Review Comments to the Author

General

Authors have adequately improved the manuscript. This manuscript can be accepted after minor revisions that I suggest authors below:

Specific

Methods

Were data collected from 2014-2023 OR from 2014-2022? Please, be consistent.

"five. . ": remove additional full stop herein.

-The data were collected from 2014-2023. The errors have now been corrected,

---

## [Editor Report · Decision Letter 3]

A dose scaling antivenin protocol in treatment of Daboia palaestinae envenomation may reduce morbidity and costs.

PONE-D-25-04569R3

Dear Dr. Zimmerman,

We’re pleased to inform you that your manuscript has been judged scientifically suitable for publication and will be formally accepted for publication once it meets all outstanding technical requirements.

Kind regards,

Timothy Omara

Academic Editor

PLOS ONE
---

## [Editor Report · Acceptance letter]

PONE-D-25-04569R3

PLOS ONE

Dear Dr. Zimmerman,

I'm pleased to inform you that your manuscript has been deemed suitable for publication in PLOS ONE. Congratulations! Your manuscript is now being handed over to our production team.

Kind regards,

on behalf of

Dr. Timothy Omara

Academic Editor

PLOS ONE